# Induction of Pro-Fibrotic CLIC4 in Dermal Fibroblasts by TGF-β/Wnt3a Is Mediated by GLI2 Upregulation

**DOI:** 10.3390/cells11030530

**Published:** 2022-02-03

**Authors:** Christopher W. Wasson, Begoña Caballero-Ruiz, Justin Gillespie, Emma Derrett-Smith, Jamel Mankouri, Christopher P. Denton, Gianluca Canettieri, Natalia A. Riobo-Del Galdo, Francesco Del Galdo

**Affiliations:** 1Leeds Institute of Rheumatic and Musculoskeletal Medicine, Faculty of Medicine and Health, University of Leeds, Leeds LS29JT, UK; j.gillespie09@gmail.com (J.G.); f.delgaldo@leeds.ac.uk (F.D.G.); 2School of Molecular and Cellular Biology, Faculty of Biological Sciences, University of Leeds, Leeds LS29JT, UK; begona.caballero@uniroma1.it (B.C.-R.); bms9jm@gmail.com (J.M.); n.a.riobo-delgaldo@leeds.ac.uk (N.A.R.-D.G.); 3Department of Molecular Medicine, Sapienza University of Rome, 00196 Rome, Italy; gianluca.canettieri@uniroma1.it; 4Centre for Rheumatology and Connective Tissue Diseases, UCL Division of Medicine, London NW32PF, UK; e.derrett-smith@ucl.ac.uk (E.D.-S.); c.denton@ucl.ac.uk (C.P.D.); 5Leeds Institute of Medical Research, Faculty of Medicine and Health, University of Leeds, Leeds LS29JT, UK; 6Scleroderma Programme, NIHR Leeds Musculoskeletal Biomedical Research Centre, Leeds LS29JT, UK

**Keywords:** fibrosis, Scleroderma, Ion channels, CLIC4, morphogens

## Abstract

Chloride intracellular channel 4 (CLIC4) is a recently discovered driver of fibroblast activation in Scleroderma (SSc) and cancer-associated fibroblasts (CAF). CLIC4 expression and activity are regulated by TGF-β signalling through the SMAD3 transcription factor. In view of the aberrant activation of canonical Wnt-3a and Hedgehog (Hh) signalling in fibrosis, we investigated their role in CLIC4 upregulation. Here, we show that TGF-β/SMAD3 co-operates with Wnt3a/β-catenin and Smoothened/GLI signalling to drive CLIC4 expression in normal dermal fibroblasts, and that the inhibition of β-catenin and GLI expression or activity abolishes TGF-β/SMAD3-dependent CLIC4 induction. We further show that the expression of the pro-fibrotic marker α-smooth muscle actin strongly correlates with CLIC4 expression in dermal fibroblasts. Further investigations revealed that the inhibition of CLIC4 reverses morphogen-dependent fibroblast activation. Our data highlights that CLIC4 is a common downstream target of TGF-β, Hh, and Wnt-3a through signalling crosstalk and we propose a potential therapeutic avenue using CLIC4 inhibitors

## 1. Introduction

Chloride intracellular channel 4 (CLIC4) can be found in soluble or membrane-associated forms and has multiple functions acting as a chloride channel and as a regulator of internal membrane and cytoskeletal dynamics. CLIC4 plays an important role in a number of processes including angiogenesis, proliferation, and inflammation [1]. CLIC4 translocates from the cytosol to the plasma membrane after serum stimulation or activation of sphingosine-1-phosphate, muscarinic M3, and lysophosphatidic acid receptors [2,3]. One function of CLIC4 is the regulation of G-protein-coupled receptor activation, leading to the activation of RhoA and Rac1 as well as the regulation of actin dynamics [3]. CLIC4 has also been reported to play an important role in cell division through acto-myosin contraction in the cleavage furrow during cytokinesis [4]. CLIC4 expression was reported to be positively regulated at the transcriptional level by functional p53 binding sites [5] and by a G-quadruplex structure in its promoter, which is important for basal expression levels [6].

Fibroblast activation, that is characterized by the expression of pro-fibrotic genes such as α-smooth muscle actin (α-SMA) and collagens, is characteristic of systemic sclerosis (SSc) and of cancer-associated fibroblasts (CAFs). CLIC4 is upregulated in CAFs [7,8] and SSc patient fibroblasts [9]. The upregulation of CLIC4 has been linked to increased fibroblast activation in response to TGF-β [7,8,9], and the increased migration, invasion, and epithelial to mesenchymal transition (EMT) of tumor cells that are associated with stromal CAFs [7]. TGF-β induces CLIC4 nuclear translocation in a Schnurri-2-dependent manner [10]. The nuclear translocation of CLIC4 is necessary for pro-fibrotic gene expression and for prolonged TGF-β signalling through preventing the de-phosphorylation of SMAD 2 and 3 [10,11]. CLIC4 is also a SMAD3 target gene in fibroblasts, exerting a positive feedback loop to amplify TGF-β signalling [10].

It is now well established that the deregulation of Wnt and Hh signalling is characteristic of fibrosis. Wnt signalling can be classified as canonical (β-catenin/TCF-dependent) or non-canonical (RhoA and calcium-dependent). Canonical Hedgehog (Hh) signalling stimulates the Smoothened (SMO)/GLI axis [12]. GLI1 and GLI2 act to drive the expression of GLI-target genes, while GLI3 serves as a transcriptional repressor. Hh signalling has also been reported to trigger the activation of RhoA/Rac1 activity and the inhibition of voltage-dependent Kv channels [13,14]. A wealth of studies indicate that Hh and Wnt signalling cascades crosstalk with the TGF-β pathway at multiple levels in positive feedback loops. Notably, SMAD3 is activated in gastric cancer cells that are stimulated with Shh [15] and the expression of GLI1 and GLI2 is upregulated by TGF-β/SMADs [16]. Furthermore, TGF-β has been shown to stimulate canonical Wnt-3a/β-catenin signalling in a p38 MAPK-dependent manner in dermal fibroblasts [17]. Concordantly, the inhibition of β-catenin blocks α-SMA expression in response to TGF-β in alveolar epithelial cells during EMT [18]. Direct interactions between SMAD3 and β-catenin have been identified at numerous promoter sites, [19] including the GLI2 promoter [20]. These findings are of interest in the context of SSc, where both Hh and Wnt-3a signalling are hyper-activated and play important roles in the activation of SSc fibroblasts [21,22].

Given the interconnection of TGF-β signalling with Hh and Wnt3a, we investigated the possibility that Hh and Wnt-3a signalling regulate CLIC4 expression in SSc fibroblasts. In this study, we show that downstream of TGF-β/SMAD3, the stimulation of canonical Wnt3a and Hh signalling pathways mediates the increase in CLIC4 expression in dermal fibroblasts and epithelial cells. In the context of SSc, we further demonstrate that CLIC4 expression can be supressed through the inhibition of β-catenin and GLI1/GLI2-dependent transcription in patient dermal fibroblasts.

## 2. Materials and Methods

### 2.1. Patient Cells

Full thickness skin biopsies were obtained from the forearms of 3 adult patients with recent onset SSc, defined as a disease duration of less than 18 months from the appearance of clinically detectable skin induration. All patients satisfied the 2013 ACR/EULAR criteria for the classification of SSc and had diffuse cutaneous clinical subset [23]. All the participants provided written informed consent to participate in the study. Informed consent procedures were approved by NRES-011NE to FDG. The fibroblasts were isolated and established as previously described [22,24]. The primary cells were immortalized using human telomerase reverse transcriptase (hTERT) to produce healthy control hTERT and SSc hTERT [22].

### 2.2. Cell Culture

The fibroblasts were maintained in Dulbecco’s modified Eagle medium (DMEM) (Gibco, Waltham, Massachusetts, USA, that was supplemented with 10% FBS (Sigma, St. Louis, Missouri, USA and penicillin-streptomycin (Sigma). The fibroblasts were treated with vehicle (DMSO, Sigma) or equal volumes of GANT61 (10 μM, Selleckchem, Houston Texas, USA, SIS3 (1–5 μM, Selleckchem), FH535 (10 μM, Tocris), NPPB (25 μM, Sigma), and IAA-94 (50 μM, Sigma) for 48 h at 37 °C and 5% CO_2_.

SW620 cells were obtained from ATCC. The cells were engineering by CRISPR/Cas9 to introduce a truncation in the PTCH1 coding sequence (PTCH1^mut^) using short guide RNA (sgRNA) targeting the region of interest (D1222fs) or an irrelevant scrambled sequence (WT PTCH1) and selected using puromycin for 72 h. The cells were sequence-verified. For the indicated treatments, the SW620 clones were incubated with GANT61 (20 µM) or SD208 (10 µM, Selleckchem) or equal volumes of DMSO for 24 h.

### 2.3. TβRIIΔk-Fib–Transgenic Mice

The generation of TβRIIΔk-fib–transgenic mice has been described previously [25]. Dermal tissue was obtained from 6–8-week-old age-matched transgenic and wild-type (WT) littermates. Strict adherence to the institutional guidelines was practiced, and full local ethics committee and Home Office approvals were obtained.

### 2.4. Immunohistochemistry

Immunohistochemistry was performed as previously described [26]. The sections were incubated in BLOXALL blocking solution (Vector Labs, San Francisco, California, USA) to quench the endogenous peroxidase activity, followed by staining with a CLIC4 antibody (Santa Cruz, Dallas, Texas, USA), visualized using an HRP conjugated mouse secondary, and counterstained with haematoxylin.

### 2.5. Morphogen Pathway Stimulation

The healthy dermal fibroblasts were serum starved for 24 h in DMEM containing 0.5% FBS and stimulated with 10 ng/mL TGF-β (R&D systems, Minneapolis, MN, USA), 100 ng/mL Wnt-3a (R&D systems), and 100 nM SMO agonist (SAG) for 24–48 h.

### 2.6. TOP Flash TCF/LEF–Firefly Luciferase Reporter

The fibroblasts were transfected with the TOP-Flash TCF/LEF–firefly luciferase reporter vector for 24 h (gift from Prof. Randall. Moon, University of Washington, Seattle, WA, USA). The pCMV-Renilla luciferase vector was co-transfected as a transfection control. These fibroblasts were then grown in serum-depleted media for 24 h prior to stimulation. The cells were then harvested and luciferase activity was measured using the dual luciferase reporter assay reagents (Promega) and detected using a luminometer (Berthold Mithras). To account for transfection efficiency, the Firefly luciferase activity was normalized to Renilla luciferase activity and expressed as the percentage change compared to the control.

### 2.7. siRNA Transfections

A pool of four siRNAs that were specific for different regions of GLI2, SMAD3, β-catenin, or a negative control scrambled siRNA (Qiagen) were transfected into fibroblasts using Lipofectamine-2000 (Thermo Fisher, Waltham, MA, USA). The fibroblasts were incubated for 48–72 h prior to harvesting.

### 2.8. Western Blotting

Total proteins were extracted from the fibroblasts in RIPA buffer and resolved by SDS-PAGE (10–15% Tris-Glycine). The proteins were transferred onto nitrocellulose membranes (Amersham biosciences) and probed with antibodies that were specific for α-SMA (Abcam), CLIC4 (Santa Cruz), GLI1 (Santa Cruz), GLI2 (R&D systems), SMAD3, β-catenin, c-Jun (Cell signalling), and β-Actin (Sigma). The immunoblots were visualized with species-specific HRP conjugated secondary antibodies (Sigma) and ECL (Thermo/Pierce) on a Biorad ChemiDoc imaging system. Densitometry analysis of the blots was performed using the ImageJ software.

### 2.9. Quantitative Real Time PCR

RNA was extracted from cells using commercial RNA extraction kits (Zymo Research). RNA (1 μg) was reverse transcribed using cDNA synthesis kits (Thermo). QRT-PCRs were performed using SyBr Green PCR kits on a Thermocycler (40 cycle programe) with primers specific for α-SMA (Forward: TGTATGTGGCTATCCAGGCG; Reverse: AGAGTCCAGCACGATGCCAG), CLIC4 (Forward: CATCCGTTTTGACTTCAGTGTTG; Reverse: AGGAGTTGTATTTAGTGTGACGA), GLI1 (Forward: GGACCTGCAGACGGTTATCC; Reverse: AGCCTCCTGGAGATGTGCAT), GLI2 (Forward: TTTATGGGCATCCTCTCTGG; Reverse: TTTTGCATTCCTTCCTGTCC), β-catenin (Forward: AAGGCTACTGTTGGATTGATTCG; Reverse: CCCTGCTCACGCAAAGGT) and GAPDH (Forward: ACCCACTCCTCCACCTTTGA; Reverse; CTGTTGCTGTAGCCAAATTCGT). The data were analysed using the ΔΔCt method using GAPDH as a housekeeping control gene.

### 2.10. Statistical Analysis

The data are presented as the mean ± standard error. Statistical analysis was performed using a two-tailed, paired Student’s *t*-test.

## 3. Results

### 3.1. Canonical Wnt and Hh Signalling Stimulate Expression of CLIC4 in Dermal Fibroblasts

To investigate the role of morphogen signalling in the regulation of CLIC4 expression, healthy human immortalized dermal fibroblasts were stimulated with TGF-β, Wnt-3a, or SAG (a small molecule SMO agonist) for 48 h. CLIC4 expression was investigated at both the protein and transcript levels (Figure 1A,B). Consistent with our previous study [7], TGF-β induced CLIC4 expression in the dermal fibroblasts at the mRNA (Figure 1A) and the protein levels (Figure 1B). In the same experimental setting, Wnt-3a and SAG increased CLIC4 expression at the mRNA and protein levels (Figure 1A,B). We also validated that increased TGF-β signalling increases CLIC4 expression in vivo using the TGF-βRIIΔK transgenic mouse model (Figure 1C). These mice exclusively expressed the TGF-βRIIΔK receptor in fibroblasts leading to increased phosphorylated SMAD 2, 3, and 4 in the skin [25,27]. A reduction in fatty tissue in the skin of the TGF-βRIIΔK mice indicated fibrosis. CLIC4 expression in the dermal fibroblasts of TGF-βRIIΔK mice skin was upregulated compared to the wild-type control animals (Figure 1C). Together these data indicate that in addition to TGF-β, Wnt, and Hh signalling activation may enhance CLIC4 expression in vitro and in vivo.

We next sought to determine if TGF-β, Wnt, and Hh signalling mediated the CLIC4 upregulation through their canonical transcription factors. To this end, we inhibited the transcriptional activity of SMAD3 (SIS3), β-catenin (FH535), or the transcription factors GLI1 and GLI2 (GANT61) with pharmacological inhibitors. SIS3 prevented TGF-β-mediated upregulation of CLIC4 in dermal fibroblasts (Figure 1D), confirming the data that were previously observed in CAFs is applicable to dermal fibroblasts. Dermal fibroblasts that were treated with Wnt-3a in combination with FH535 suppressed the ability of Wnt-3a to induce CLIC4 upregulation at the protein level and reduced expression of CLIC4 below the basal level (Figure 1E). Similarly, GANT61 prevented the SAG-mediated increase in CLIC4 (Figure 1F). Densitometry analysis confirmed a statically significant reversal of Wnt3a- and SAG-mediated CLIC4 expression when the canonical transcription factors were inhibited (Figure 1G). Strikingly, α-SMA expression closely correlated with CLIC4 expression levels in all the conditions, confirming a previous report that CLIC4 silencing abolished α-SMA expression in fibroblasts [7]. These data suggest that, in addition to SMAD3, β-catenin and the GLI transcription factors contribute to CLIC4 expression in dermal fibroblasts.

### 3.2. Wnt3a and Hh Signalling Cooperate with SMAD3 to Enhance CLIC4 Expression

The evidence above shows that the stimulation of the canonical Wnt and Hh signalling pathways can induce the expression of CLIC4 in dermal fibroblasts, in addition to the previously described role for the TGF-β signalling pathway [9]. These three pathways may regulate CLIC4 independently or in co-operation. As discussed above, these three pathways interact in a number of cell types. To establish a relationship among TGF-β and Wnt signalling in dermal fibroblasts, we investigated the effect of SMAD3 inhibition with siRNA or a small molecule inhibitor on Wnt/β-catenin-dependent transcription, as determined using a TOP Flash-luciferase reporter. The TOP Flash-luciferase construct has a TCF binding site that is upstream of the firefly luciferase gene. TCF binds to β-catenin and recruits β-catenin to specific promoters. As seen in Figure 2A, a reduction in the SMAD3 protein level led to an inhibition of the TOP Flash-luciferase activity in response to Wnt3a compared to scramble control. In support, the SMAD3 inhibitor SIS3 reduced Wnt3a-mediated stimulation of TOP Flash-luciferase activity in a dose-dependent manner (Figure 2B). Stimulation of healthy fibroblasts with TGF-β increased the β-catenin protein levels (Figure 2C), providing an explanation of the enhancement of canonical Wnt signalling by TGF-β/SMAD3 signalling.

Further analysis of interplay between the pathways revealed a relationship between the Wnt and Hh signalling pathways in dermal fibroblasts. The stimulation of dermal fibroblasts with Wnt3a increased the GLI2 transcript levels in a β-catenin-dependent manner as GLI2 upregulation was inhibited by FH535 treatment (Figure 2D). This suggests that GLI2 is a transcriptional target of β-catenin in dermal fibroblasts. Interestingly, β-catenin is an important intermediator for TGF-β-mediated stimulation of GLI2 expression. Consistent with previous findings [28], TGF-β enhances GLI2 expression in dermal fibroblasts. This study revealed this is reversed when β-catenin is inhibited by FH535 (Figure 2E,F), highlighting a relationship among TGF-β, Wnt, and Hh signalling in dermal fibroblasts. TGF-β/SMAD3 can stimulate the expression of the Hh transcription factors through β-catenin. Finally, we revealed that GLI2 can stimulate the expression of β-catenin. As shown in Figure 2G,H, the stimulation of the Hh pathway with SAG leads to an increased expression of the β-catenin protein and transcript levels, which were prevented when the GLI transcription factors were inhibited with GANT61. Therefore, high levels of GLI1/2 in dermal fibroblasts may also enhance Wnt3a signalling.

We described the above relationships between the TGF-β, Wnt, and Hh signalling pathways in dermal fibroblasts. Next, we sought to investigate the effects of this relationship on CLIC4 and α-SMA expression in dermal fibroblasts. To investigate if CLIC4 upregulation in response to Wnt3a and Hh pathway activation is also potentiated by SMAD3, we treated healthy dermal fibroblasts with Wnt3a or SAG in combination with SIS3. Interestingly, SIS3 blocked both Wnt3a- and SAG-induced CLIC4 and α-SMA upregulation (Figure 2I). Finally, we investigated whether β-catenin and GLI2 are required for induction of CLIC4 by TGF-β. The healthy dermal fibroblasts were stimulated with TGF-β in combination with the SMAD3 inhibitor SIS3, the β-catenin inhibitor FH535, or the GLI1/2 inhibitor GANT61. Remarkably, blocking β-catenin or GLI1/2 inhibited TGF-β-mediated CLIC4 and α-SMA expression to a greater extent than the SMAD3 inhibitor (Figure 2J). Taken together, these results suggest the morphogens can activate one another which, in turn, stimulates CLIC4 and α-SMA expression (Figure 2K).

### 3.3. Regulation of CLIC4 Expression by SMADs and GLI Signalling Crosstalk Is Not Limited to Fibroblasts

In a parallel approach, we investigated if the observed regulation of CLIC4 by crosstalk of TGF-β and Hh signalling is specific to dermal fibroblasts or a widespread phenomenon. To this end, we used SW620 cells, a colon cancer cell line without mutations in the Hh pathway in which we engineered an oncogenic PTCH1 mutation using CRISPR/Cas9 technology. RNA-seq analysis of TGF-β and Hh signalling components revealed that PTCH1^mut^ cells expressed higher mRNA levels of SMAD2, SMAD3, and SMAD4, higher levels of SMO; lower levels of GLI3; and higher levels of CLIC4 (Figure 3A). The upregulation of SMAD2/3/4 and CLIC4 was confirmed by Western blotting (Figure 3B). The higher levels of SMAD2/3/4 suggest an upregulation of SMAD-dependent signalling, while the upregulation of SMO together with a significant reduction in GLI3 expression (Hh repressor), suggest increased GLI transcriptional activity. As predicted, PTCH1^mut^ cells had increased GLI1 mRNA (Figure 3C), a hallmark of increased canonical Hh signalling, and increased CLIC4 mRNA (Figure 3D) compared to the control cells, which were also evident at the protein level (Figure 4E). The treatment of PTCH1^mut^ cells with GANT61 reduced both GLI1 and CLIC4 expression (Figure 3C–E). In the same experimental setting, we analysed the effect of the TGF-β receptor inhibitor SD208. SD208 reduced GLI1 and CLIC4 mRNA and protein expression in PTCH1^mut^ cells (Figure 3C–E). Taken together, these data support a widespread crosstalk of Hh and TGF-β signalling in different cell types that play an essential role in regulating CLIC4 expression.

### 3.4. β-Catenin Contributes to CLIC4 Overexpression in SSc Fibroblasts

We have previously reported that CLIC4 is overexpressed in SSc patient fibroblasts through the activity of the SMAD3 transcription factors [9]. Since Wnt/β-catenin signalling is dysregulated in SSc fibroblasts [22], we set out to determine if Wnt/β-catenin signalling contributes to the high expression of CLIC4 in SSc fibroblasts. Healthy and SSc immortalized fibroblasts were treated with the β-catenin inhibitor FH535 in the absence of exogenous TGF-β. The FH535 treatment partially reduced the transcript and protein CLIC4 levels in SSc fibroblasts, but had no effect on the basal CLIC4 expression in healthy fibroblasts (Figure 4A,C). Similarly, FH535 reduced α-SMA levels only in the SSc fibroblasts (Figure 4B,C) further confirming the previous reports that CLIC4 is necessary for α-SMA expression. We ruled out the possibility that the results that were seen with FH535 were due to off-target effects by silencing β-catenin by siRNA. The silencing of β-catenin caused a reduction in CLIC4 levels in the SSc fibroblasts compared to the scrambled siRNA control (Figure 4D). These data suggest that enhanced β-catenin signalling contributes to CLIC4 upregulation in SSc.

### 3.5. Enhanced CLIC4 Levels in SSc Fibroblasts Are Mediated by GLI2 Expression

Dysregulated Hh signalling is a characteristic feature of SSc fibroblasts [21]. In view of the involvement of GLI2 in CLIC4 induction in healthy fibroblasts (Figure 1), we hypothesized that GLI2 might be required for the CLIC4 upregulation that is observed in SSc fibroblasts. To test this hypothesis, healthy and SSc immortalized fibroblasts were treated with the GLI1/GLI2 inhibitor GANT61. GANT61 treatment reduced the CLIC4 transcript and protein levels in the SSc fibroblasts to levels of healthy fibroblasts, but had no effect on the CLIC4 expression in healthy fibroblasts (Figure 5A,B). These data suggest a role for GLI transcription factors in the overexpression of CLIC4 in SSc fibroblasts.

GLI2 is the main mediator of the canonical Hh pathway in response to Shh or to a direct SMO agonist, while GLI1 is a transcriptional target of GLI2 [12]. We investigated the specific contribution of GLI2 using siRNA silencing approaches in healthy and SSc fibroblasts. GLI2 silencing was confirmed by qPCR and Western blot analysis (Figure 5C,E). GLI2 siRNA significantly reduced the expression of CLIC4 in SSc fibroblasts, but not in the healthy cells (Figure 4D,E). These data confirmed a key role for GLI2 in the overexpression of CLIC4 that is observed in SSc fibroblasts. In addition, the inhibition of GLI1/2 in SSc fibroblasts blocked β-catenin expression (Figure 5B,E). This further validates the influence GLI1/2 has on β-catenin that was first observed in Figure 2G,H.

### 3.6. Chloride Channel Inhibition Suggests a Role for CLIC4 in TGF-β, Wnt3a, and Hh-Dependent Fibroblast Activation

TGF-β, Wnt3a, and Hh signalling pathways are important regulators of dermal fibroblasts activation [17,21,22] and inhibition of chloride channels with the inhibitors NPPB and IAA-94 led to reduced fibroblast activation marker expression [9]. Therefore, CLIC4 maybe an important downstream intermediate of TGF-β, Wnt3a, and Hh fibroblast activation. To test this hypothesis, we stimulated healthy dermal fibroblasts with TGF-β, Wnt3a, and SAG in combination with the inhibitors NPPB and IAA-94, which are known to target CLIC4 among other chloride channels [29]. Both inhibitors reduced α-SMA expression in response to the morphogens (Figure 6A–C). Given that genetic ablation of CLIC4 blocked TGF-β fibroblast activation in CAFs, it is likely that the inhibitors effect is due to a reduction in CLIC4 function [7]. In addition, the inhibitors also reduced TGF-β-dependent upregulation of β-catenin and GLI2 expression (Figure 6A). This suggests that the inhibition of CLIC4 function alone or in cooperation with another chloride channel disrupts the crosstalk between TGF-β, Wnt, and Hh signalling in dermal fibroblasts. Further analysis suggested that CLIC4 activity is important for Wnt3a signalling. Both inhibitors prevented β-catenin and c-jun (β-catenin target gene [30]) expression in response to Wnt-3a (Figure 6B). Finally, the chloride channel inhibitors reduced β-catenin upregulation by the Hh pathway (Figure 6C), further strengthening the hypothesis that GLI transcriptional activity modulates the β-catenin levels (as shown in Figure 2G–H). The inhibitors had no effect on the SMAD3 levels in response to Wnt3a and SAG. This suggests β-catenin and GLI1/2 do not regulate SMAD3 expression in fibroblasts. Excitingly, we observed a reduced expression of β-catenin and GLI2 in SSc fibroblasts that were treated with the inhibitors (Figure 6D). This suggests that CLIC4 might fine-tune the strength of endogenous Wnt3a and Hh signalling in SSc fibroblasts as well as in ligand-stimulated healthy fibroblasts.

## 4. Discussion

CLIC4 is a TFG-β target gene and a key mediator of SMAD2/3-mediated transcription [7]. CLIC4 expression is upregulated by TGF-β in SSc fibroblasts [9]. In our previous study, we demonstrated that the SSc-myofibroblast phenotype requires CLIC4. In this study, we show that, in addition to its regulation by TGF-β, CLIC4 expression requires the activation of canonical Wnt and Hh signalling, acting through their transcription factors β-catenin and GLI1/GLI2, respectively. The activation of these pathways in SSc dermal fibroblasts, in part as a result of excessive TGF-β/SMAD signalling, is necessary for CLIC4 upregulation and the consequent α-SMA expression.

The mechanistic basis of the functional cooperation among SMAD3, β-catenin, and GLI1/GLI2 is not fully understood. SMAD3 has been shown to induce nuclear translocation of β-catenin in mesenchymal stem cells [31] and upregulates some Wnt family members in vascular smooth muscle cells [32]. SMAD2/3 and β-catenin have been reported to cooperate to stimulate GLI2 transcription by binding to specific elements in the GLI2 promoter [20]. The data that are presented in this study suggests that TGF-β in SSc fibroblasts increases Wnt/β-catenin signalling, and that both SMAD3 and/or β-catenin increase the GLI2 expression. This is interesting because previous studies in SSc fibroblasts have shown Wnt3a/β-catenin is an important mediator of TGF-β-mediated SSc fibroblast activation [15] and TGF-β induces GLI2 expression in SSc fibroblasts [28]. Our study shows Wnt3a/β-catenin is an important intermediate for TGF-β-mediated GLI2 expression. This, in turn, drives CLIC4 expression. This is supported by the repression of CLIC4 expression by GANT61 and GLI2 siRNA in SSc fibroblasts. The model that is described above also explains the findings in another cell type, in which the dysregulation of Hh and TGF-β signalling by an oncogenic mutation in PTCH1 leads to CLIC4 induction.

This relationship between the signalling pathways is observed in vivo. CLIC4 expression is increased in the dermis of the TGF-βRIIΔK transgenic mice. This suggests that the increased activation of the TGF-β signalling pathway drives the increased CLIC4 expression in vivo. Interestingly, previous studies showed a decreased expression of Axin2 in the dermis of these mice [22]. Axin2 is one of the major components of the β-catenin destruction complex. Therefore, it is possible that elevated levels of β-catenin may have a role in the increased expression of CLIC4 in vivo.

The use of pharmacological inhibitors is associated with potential non-specific effects or cytotoxicity. However, GANT61 and GLI2 siRNA as well as FH535 and β-catenin siRNA exerted a comparable inhibition of CLIC4 expression. The less profound effect of the SMAD3 inhibitor SIS3 on CLIC4 induction by TGF-β could be explained by the redundant function of SMAD2, which is not targeted by SIS3. Therefore, our findings strongly support a sequence of signalling events that cooperate to drive CLIC4 expression.

Some findings suggest that the crosstalk between Hh and TGF-β signalling can be bidirectional. In PTCH1^mut^ cells, dysregulated Hh signalling is associated with the upregulation of SMAD2, 3, and 4 mRNA, as was also shown for SMAD3 in gastric cancers [15]. We also observed a bidirectional relationship between the Hh and Wnt3a signalling in fibroblasts. β-catenin expression levels were increased in dermal fibroblasts that were stimulated with SAG and this was reversed when the GLI transcription factors were inhibited. These could, in turn, act as a positive feedback loop to further increase and/or maintain CLIC4 levels.

β-Catenin and the GLI transcription factors may also drive CLIC4 expression in CAFs. Previous studies have shown both Hh and Wnt/β-catenin signalling contribute to the phenotypes of CAFs. For example, pancreatic-derived CAFs show elevated levels of SMO and GLI1 expression [33], whilst melanoma-associated CAFs exhibit reduced extracellular matrix production and paracrine signalling when β-catenin is inhibited [34]. Therefore, in addition to the well characterized role of TGF-β, Wnt-3a and Hh pathways may further drive CLIC4 expression in CAFs as in SSc fibroblasts and colon cancer cells that are shown here.

The data that are presented in this study further highlights the importance of CLIC4 in mediating dermal fibroblast activation. The inhibition of CLIC4 with small molecule inhibitors blocked morphogen-mediated fibroblast activation (Figure 6). CLIC4 is known to regulate TGF-β signalling in CAFs but this is the first example of CLIC4’s ability to regulate Wnt3a and Hh signalling in any cell type. Future work is required to determine if CLIC4 is required to maintain SMAD2–3 activation in SSc fibroblasts as previously shown in CAFs [10] and to determine the specific mechanisms behind CLIC4 ability to regulate Wnt-3a and Hh signalling (Figure 7).

Overall, we have shown that CLIC4 is a major cellular factor that is involved in SSc pathophysiology. SSc disease progression is initiated, in part, by dermal fibroblast activation through morphogen signalling. Here we show that CLIC4 is a downstream target of morphogen signalling and contributes to fibroblast activation. These data support the notion of CLIC4 as a therapeutic target of fibrosis in SSc.

## Figures and Tables

**Figure 1 cells-11-00530-f001:**
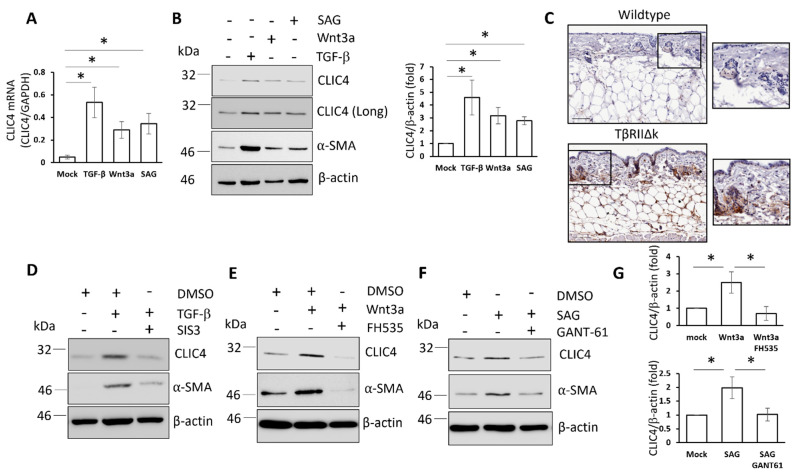
CLIC4 expression is regulated by a number of morphogens in dermal fibroblasts. The healthy dermal fibroblasts were grown in serum-depleted media and stimulated with TGF-β, Wnt3a, and Smoothened agonist (SAG) for 48 h. CLIC4 transcript (**A**) and protein levels (**B**,**C**) were assessed. (**C**) Immunohistochemistry analysis for CLIC4 expression (Brown) in skin sections from wild-type and TβRIIΔK-fib transgenic mice. The scale bars represent 100 μm. Healthy dermal fibroblasts that were stimulated with TGF-β (**D**), Wnt3a (**E**), or SAG (**F**) for 48 h in the absence or presence of 1 μM SIS3 (**D**), 10 μM FH535 (**E**), or 10 μM GANT61 (**F**). CLIC4 and α-SMA protein levels were assessed. (**G**) Densitometry analysis of CLIC4 blots from (**E**) and (**F**). The graphs represent the means +/− standard error for three independent experiments. * *p* < 0.05.

**Figure 2 cells-11-00530-f002:**
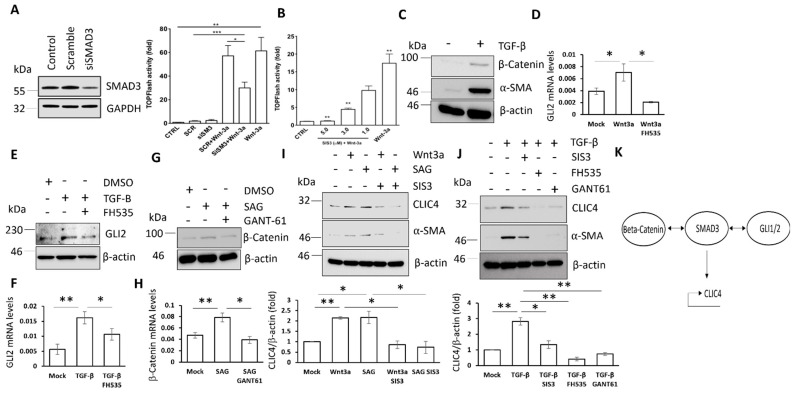
Wnt/Hh/TGF-β cross talk is important for CLIC4 expression in dermal fibroblasts. Healthy dermal fibroblasts that were treated with SMAD3/scramble siRNA (**A**,**B**) or SIS3 (**C**), followed by transfection with the TOP-Flash-Firefly and CMV-Renilla luciferase reporters prior to treatment with Wnt3a. The relative TOP-Flash reporter activity was determined by the dual luciferase assay. (**C**) β-catenin and α-SMA protein levels from TGF-β-stimulated fibroblasts. (**D**) GLI2 transcript from Wnt3a/FH535-treated fibroblasts. GLI2 protein (**E**) and transcript (**F**) from TGF-β/FH535-treated fibroblasts. β-catenin protein (**G**) and transcript (**H**) from SAG/GANT61-treated fibroblasts. (**I**) CLIC4 and α-SMA protein from Wnt3a/SAG/SIS3-treated fibroblasts. (**J**) CLIC4 and α-SMA protein from TGF-β/SIS3/FH535/GANT61-treated fibroblasts (**K**) Schematic of the hierarchical relationship. The graphs represent the means +/− standard error for three independent experiments. * *p* < 0.05, ** *p* < 0.01, *** *p* < 0.001.

**Figure 3 cells-11-00530-f003:**
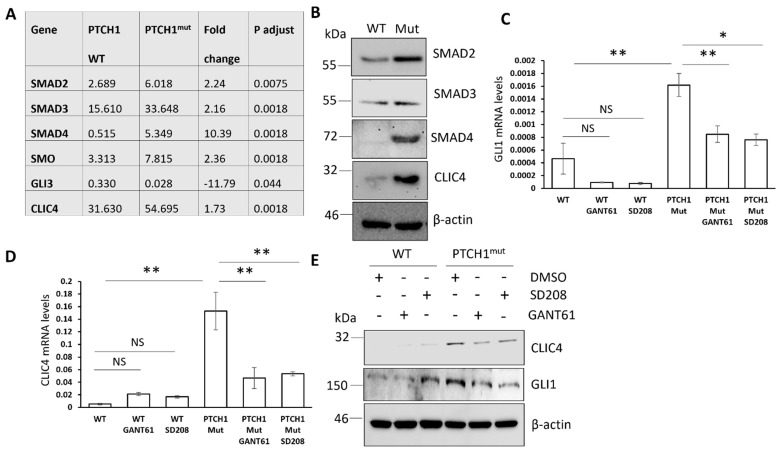
The regulation of CLIC4 expression by TGF-β and Hh signaling is conserved in other cell types. RNA and protein were extracted from Isogenic SW620 cells expressing endogenous WT PTCH1 or mutated PTCH1 (PTCH1^mut^). RNA-seq analysis was performed on the RNA. (**A**) The table represents the fold change of genes relating to the TGF-β and Hh signalling pathways. (**B**) SMAD2, 3, 4, and CLIC4 protein levels were analysed by Western blot. WT PTCH1 or mutated PTCH1 (PTCH1^mut^) were treated with 20 μM GANT61, 10 μM SD208, or DMSO for 24 h. GLI1 (**C**) and CLIC4 (**D**) transcript levels and protein levels (**E**) were assessed. The graphs represent the mean +/− standard error of three independent experiments. * *p* < 0.05, ** *p* < 0.01.

**Figure 4 cells-11-00530-f004:**
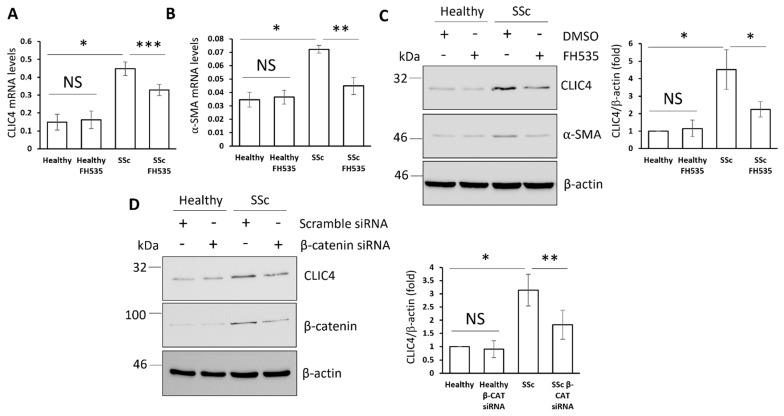
β-Catenin contributes to the enhanced CLIC4 expression in SSc fibroblasts. RNA and protein were extracted from healthy and SSc dermal fibroblasts that were treated with 10 μM FH535 or DMSO (vehicle) for 48 h. CLIC4 (**A**), α-SMA (**B**) transcript, CLIC4, and α-SMA protein levels (**C**) were assessed. The healthy and SSc fibroblasts were transfected with siRNA that was specific to β-catenin or scramble control siRNA. (**D**) CLIC4 and β-catenin protein levels were assessed by Western blot. β-actin served as a loading control. The graphs represent the mean +/− standard error for three independent experiments using three different patient cell lines. * *p* < 0.05, ** *p* < 0.01, *** *p* < 0.001.

**Figure 5 cells-11-00530-f005:**
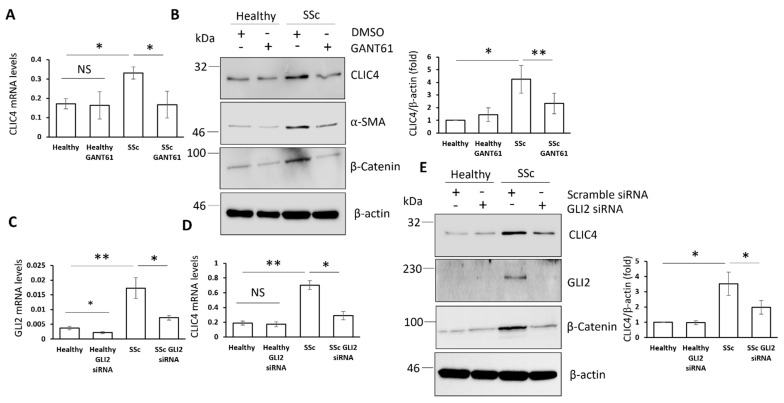
GLI2 is an important mediator of enhanced CLIC4 levels in SSc fibroblasts. CLIC4 transcript levels (**A**) and protein levels (**B**) from healthy and SSc fibroblasts that were treated or not with 10 μM GANT61. The healthy and SSc fibroblasts were transfected with siRNA that was specific to GLI2 or scramble control siRNA. (**C**) The GLI2 transcript levels were assessed by q-RT-PCR. (**D**) CLIC4 mRNA levels were quantified in the same samples from (**C**). (**E**) CLIC4, GLI2, and β-catenin protein levels from healthy and SSc fibroblasts that were treated with GLI2 or scrambled siRNA were assessed by Western blot. β-actin served as a loading control. The graphs represent the mean +/− standard error for three independent experiments using three different patient cell lines. * *p* < 0.05, ** *p* < 0.01.

**Figure 6 cells-11-00530-f006:**
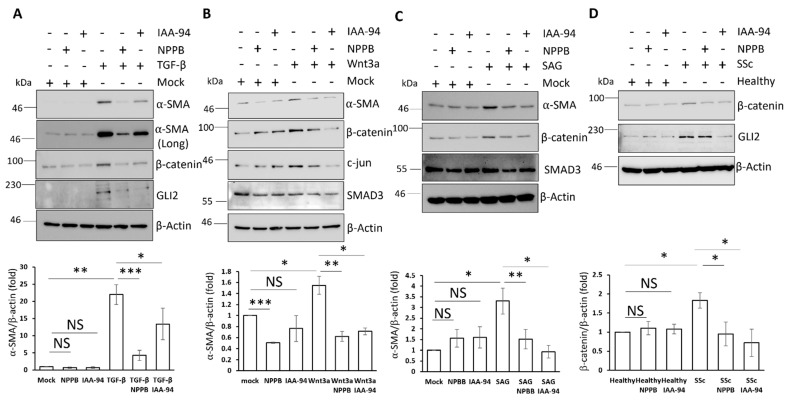
The inhibition of CLIC4 blocks Wnt3a-/Hh-/TGF-β-mediated fibroblast activation. Healthy dermal fibroblasts were stimulated with TGF-β (**A**), Wnt3a (**B**), and SAG (**C**) in the presence and absence of the chloride channel inhibitors NPPB and IAA-94 for 48 h. The α-SMA, β-catenin, SMAD3, and GLI2 protein levels were analysed by Western blot. β-actin served as a loading control. (**D**) The healthy and SSc patient fibroblasts were treated with NPPB and IAA-94 for 48 h. The β-catenin and GLI2 protein levels were analysed by Western blot. β-actin served as a loading control. The graphs represent the mean +/− standard error for three independent experiments. * *p* < 0.05, ** *p* < 0.01, *** *p* < 0.001.

**Figure 7 cells-11-00530-f007:**
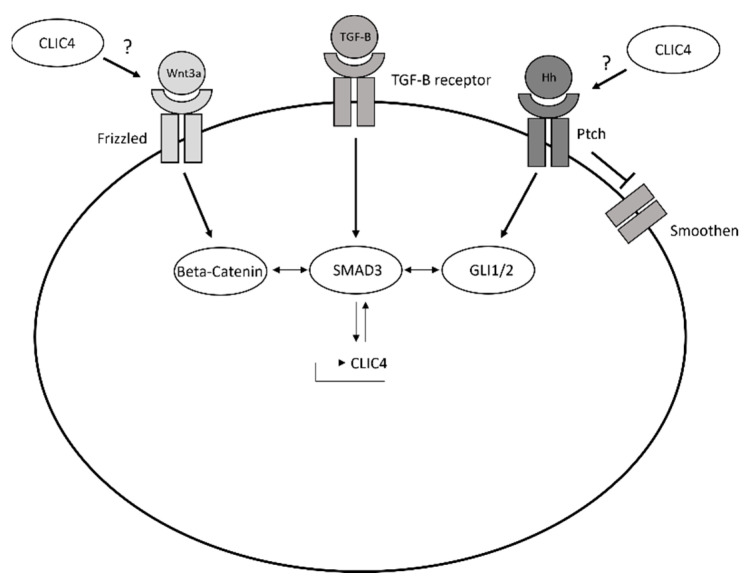
The schematic of the role of CLIC4 in morphogen-mediated fibroblast activation.

## Data Availability

All data generated or analysed during this study are included in the published article. Datasets are available from the corresponding author upon reasonable request.

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
