# Peer review of "Induction of Pro-Fibrotic CLIC4 in Dermal Fibroblasts by TGF-β/Wnt3a Is Mediated by GLI2 Upregulation"

_cells, 2022, doi:10.3390/cells11030530_

Round 1
Reviewer 1 Report
Wasson et al have reported previously on the upregulation of CLIC4 in the dermis of patients with scleroderma and in this report they attempt to dissect the pathways leading to CLIC4 upregulation. Using mainly immortalized human normal dermal fibroblasts and those from scleroderma patients, they present strong evidence for the known regulation of CLIC4 by TGFbeta and through stepwise use of siRNA and pharmacological inhibitors invoke wnt and hedgehog co-stimulation through beta-catenin and gli transcription factors. The results suggest these factors are associated with the elevated CLIC4 and alpha SMA in scleroderma but lack direct evidence of promoter binding assays. While the data support the conclusions as far as they have been taken, it is a reach to draw the conclusions that this is a linear transcriptional relationship and thus, their schematic assumptions should be toned down until more is known.
Major issues:
- The authors should include somewhere (intro, discussion) the presentation of factors that are known to regulate CLIC4 transcription such as p53 and G-quadraplex formation. Along these same lines, the authors should explore the database of potential transcription factors binding to the CLIC4 promoter to see if the transcription factors they report have consensus binding sites. That would strengthen their data if present or possibly indicate that the pathways they enumerate have indirect effects.
- The number of independent samples of normal and scleroderma fibroblasts tested is not clear from the text or figure legends. Does 3 independent experiments indicate the same patient sample used 3 times or 3 different patient samples. In fact, it would be very helpful to have some of the experiments in Figure 5 performed on 3 independent samples to ensure that all patient fibroblasts respond similarly.
- The data in Figure 6 is weak and does not stand alone. The inhibitors used are non-specific and are general ion channel inhibitors. The evidence that CLIC4 is an ion channel is very weak and most likely it does not work through channel activity but more likely is regulating cellular activity through its known redox regulating enzymatic activity or some other mechanism. These channel inhibitors are probably affecting something else. To conclude that CLIC4 is having the feedback activity proposed, the experiment would have to use siRNA, antisense reagents or CRISPR knockout. Until that is done the authors cannot conclude that CLIC4 channel activity, if it exists at all, has anything to do with the results.
Minor issues: throughout there are mistakes with Greek letters missing or references in two different formats. All minor stuff.
In conclusion this is an important study in an important disease with new insights that might have therapeutic implications with stronger experimental support. The authors should be encouraged to expand the data to provide more solid support for their conclusions.
Author Response
Reviewer 1
Wasson et al have reported previously on the upregulation of CLIC4 in the dermis of patients with scleroderma and in this report they attempt to dissect the pathways leading to CLIC4 upregulation. Using mainly immortalized human normal dermal fibroblasts and those from scleroderma patients, they present strong evidence for the known regulation of CLIC4 by TGFbeta and through stepwise use of siRNA and pharmacological inhibitors invoke wnt and hedgehog co-stimulation through beta-catenin and gli transcription factors. The results suggest these factors are associated with the elevated CLIC4 and alpha SMA in scleroderma but lack direct evidence of promoter binding assays. While the data support the conclusions as far as they have been taken, it is a reach to draw the conclusions that this is a linear transcriptional relationship and thus, their schematic assumptions should be toned down until more is known.
We thank the reviewer for their comments on our manuscript. We agree that the linear transcriptional relationship between the transcription factors cannot be concluded from the current data and we have adjusted the schematic and associated text to suggest co-operation between the factors in regulating CLIC4 transcription instead of a hierarchal relationship.
Major issues:
- The authors should include somewhere (intro, discussion) the presentation of factors that are known to regulate CLIC4 transcription such as p53 and G-quadraplex formation. Along these same lines, the authors should explore the database of potential transcription factors binding to the CLIC4 promoter to see if the transcription factors they report have consensus binding sites. That would strengthen their data if present or possibly indicate that the pathways they enumerate have indirect effects.
We have now discussed the roles of p53 and G-quadraplex formation on CLIC4 transcription in the introduction (citing the relevant studies).
Preliminary analysis of the CLIC4 promoter suggests a partial consensus sequences for GLI in the -3 kbp fragment upstream from the start site but that this needs experimental verification in future work before we can definitively state the GLI transcription factors binds to the promoter.
- The number of independent samples of normal and scleroderma fibroblasts tested is not clear from the text or figure legends. Does 3 independent experiments indicate the same patient sample used 3 times or 3 different patient samples. In fact, it would be very helpful to have some of the experiments in Figure 5 performed on 3 independent samples to ensure that all patient fibroblasts respond similarly.
Experiments were performed in three different patient cell lines including the experiments in figure 5. This has now been made clear in the manuscript.
- The data in Figure 6 is weak and does not stand alone. The inhibitors used are non-specific and are general ion channel inhibitors. The evidence that CLIC4 is an ion channel is very weak and most likely it does not work through channel activity but more likely is regulating cellular activity through its known redox regulating enzymatic activity or some other mechanism. These channel inhibitors are probably affecting something else. To conclude that CLIC4 is having the feedback activity proposed, the experiment would have to use siRNA, antisense reagents or CRISPR knockout. Until that is done the authors cannot conclude that CLIC4 channel activity, if it exists at all, has anything to do with the results.
We agree that the evidence in the literature for CLIC4 ion channel activity is weak and requires further work. The inhibitors used in this study have been shown in a number of studies to block CLIC4 functions (Proutki et al 2002, Domingo-Fernandez 2017). Therefore we believe our data shows blocking CLIC4 function with these inhibitors affects the morphogen signalling pathways. But we have re-worded the manuscript to describe that inhibition of chloride channel activity can reverse morphogen mediated fibroblast activation.
Minor issues: throughout there are mistakes with Greek letters missing or references in two different formats. All minor stuff.
These mistakes have been identified in the text and have now been corrected.
In conclusion this is an important study in an important disease with new insights that might have therapeutic implications with stronger experimental support. The authors should be encouraged to expand the data to provide more solid support for their conclusions.
Again we thank the reviewer for their comments. We intend to further explore the role of CLIC4 in the disease and in morphogen signalling in future grants and manuscripts.
Reviewer 2 Report
Introduction
- “Concordantly, inhibition of b-catenin blocks a-SMA expression in response to TGF beta in alveolar epithelial cells (16).” Is that during EMT? Please expand on this since a-SMA is usually not expressed by epithelial cells
Methods/Results
- “Primary cells were immortalized using human telomerase 83 reverse transcriptase (hTERT) to produce healthy control hTERT and SSc hTERT.” Results in this study could have been affected by immortalization. This needs to be discussed in the discussion section. Throughout the manuscript when these cells were used it should be specified they were “immortalized dermal fibroblasts.” Were the SSc and healthy fibroblasts that were compared both immortalized?
- Specify what controls were used in IHC. Pre-absorption, non-specific isotypic antibody or what?
- More specifics. For example, how many cycles? Other details so the results can be repeated by other investigators.
Discussion
- CLIC4 expression is upregulated by TGF-b SSc fibroblasts. Should be CLIC4 expression is upregulated by TGF-b in SSc fibroblasts.
- “The use of pharmacological inhibitors is subjected to potential non-specific effects or cytotoxicity.” Better as “The use of pharmacological inhibitors is associated with potential non-specific effects or cytotoxicity.”
- So, what does this have to do with the pathophysiology of scleroderma? More discussion of the relevance of the findings to those interested in the disease.
Author Response
Reviewer 2
Introduction
- “Concordantly, inhibition of b-catenin blocks a-SMA expression in response to TGF beta in alveolar epithelial cells (16).” Is that during EMT? Please expand on this since a-SMA is usually not expressed by epithelial cells.
This occurs during EMT and we have now clarified this in the introduction
Methods/Results
- “Primary cells were immortalized using human telomerase 83 reverse transcriptase (hTERT) to produce healthy control hTERT and SSc hTERT.” Results in this study could have been affected by immortalization. This needs to be discussed in the discussion section. Throughout the manuscript when these cells were used it should be specified they were “immortalized dermal fibroblasts.” Were the SSc and healthy fibroblasts that were compared both immortalized?
Both our healthy and SSc patient fibroblasts were immortalized. We have now stated throughout the manuscript these were immortalized fibroblasts.
- Specify what controls were used in IHC. Pre-absorption, non-specific isotypic antibody or what?
Prior to staining we quenched endogenous peroxidase activity with the BLOXALL blocking solution. This has now been stated in the methods
- More specifics. For example, how many cycles? Other details so the results can be repeated by other investigators.
We have now added additional information in the methods
Discussion
- CLIC4 expression is upregulated by TGF-b SSc fibroblasts. Should be CLIC4 expression is upregulated by TGF-b in SSc fibroblasts.
This has been altered in the text
- “The use of pharmacological inhibitors is subjected to potential non-specific effects or cytotoxicity.” Better as “The use of pharmacological inhibitors is associated with potential non-specific effects or cytotoxicity.”
This has been altered in the text
- So, what does this have to do with the pathophysiology of scleroderma? More discussion of the relevance of the findings to those interested in the disease.
This is a valid point to provide more context to these finding in the disease. In the discussion we have now added a section explain the context of these finding in relations to SSc
Round 2
Reviewer 1 Report
Manuscript is improved. There is still some room to tell more about CLIC4 and its potential role in the disease such as its redox activity that could be important but not discussed. Figure 6 is improved but a knockdown would be more definitive. The removal of the hierarchy proposal is a big improvement since the original conclusion was unlikely to be correct.
Reviewer 2 Report
Thank you for responding to the review, which has improved the manuscript substantially.